# Explaining COVID-19 mortality among immigrants in Sweden from a social determinants of health perspective (COVIS): protocol for a national register-based observational study

Sol Pia Juárez [1,2] Helena Honkaniemi [1,2] Siddartha Aradhya [3]
Enrico Debiasi [1,2] Srinivasa Vittal Katikireddi [4] Agneta F Cederström [1,2]
Eleonora Mussino [3] Mikael Rostila [1,2]

[1]Department of Public Health Sciences, Stockholm University, Stockholm, Sweden
[2]Centre for Health Equity Studies (CHESS), Stockholm University/ Karolinska Institutet, Stockholm, Sweden
[3]Stockholm University Demography Unit, Stockholm University, Stockholm, Sweden
[4]MRC/CSO Social & Public Health Sciences Unit, University of Glasgow, Glasgow, UK

**Correspondence to**
Dr Sol Pia Juárez;
sol.juarez@su.se

## ABSTRACT

**Introduction** Adopting a social determinants of health perspective, this project aims to study how disproportionate COVID-19 mortality among immigrants in Sweden is associated with social factors operating through differential exposure to the virus (eg, by being more likely to work in high-exposure occupations) and differential effects of infection arising from socially patterned, pre-existing health conditions, differential healthcare seeking and inequitable healthcare provision.

**Methods and analysis** This observational study will use health (eg, hospitalisations, deaths) and sociodemographic information (eg, occupation, income, social benefits) from Swedish national registers linked using unique identity numbers. The study population includes all adults registered in Sweden in the year before the start of the pandemic (2019), as well as individuals who immigrated to Sweden or turned 18 years of age after the start of the pandemic (2020). Our analyses will primarily cover the period from 31 January 2020 to 31 December 2022, with updates depending on the progression of the pandemic. We will evaluate COVID-19 mortality differences between foreign-born and Swedish-born individuals by examining each mechanism (differential exposure and effects) separately, while considering potential effect modification by country of birth and socioeconomic factors. Planned statistical modelling techniques include mediation analyses, multilevel models, Poisson regression and event history analyses.

**Ethics and dissemination** This project has been granted all necessary ethical permissions from the Swedish Ethical Review Authority (Dnr 2022-0048-01) for accessing and analysing deidentified data. The final outputs will primarily be disseminated as scientific articles published in open-access peer-reviewed international journals, as well as press releases and policy briefs.

## INTRODUCTION

SARS-CoV-2 and its related disease (COVID-19) represent an unprecedented public health concern, with 760 million cases and

## STRENGTHS AND LIMITATIONS OF THIS STUDY

⇒ The study will offer a thorough empirical evaluation of how social conditions shape group risks in the context of a pandemic giving rise to native-immigrant inequalities in COVID-19 mortality.

⇒ The study will offer a comprehensive understanding of native-immigrant inequalities across the COVID-19 disease pathway (ie, positive test, hospitalisation, intensive care unit admission and death).

⇒ This study will be restricted to social information available in administrative registers, therefore, it will not be able to account for self-reported racism and discrimination.

⇒ Analyses will not be able to include undocumented immigrants, a highly vulnerable population.

6.8 million deaths internationally as of March 2023.[1] Since the early stages of the pandemic, immigrants and ethnic minorities have been disproportionately affected by COVID-19. Notably, evidence from the UK and USA has found that ethnic minorities are not only at a higher risk of being infected by SARS-CoV-2, but also of developing severe complications and dying from COVID-19, relative to majority ethnic groups.[2–5] A systematic review and meta-analysis has confirmed this pattern for immigrants in most high-income countries.[6]

Sweden has been no exception. Despite its generous immigrant integration approaches and guarantee of equal access to healthcare and social protection for all documented immigrants, immigrants from low-income to high-income countries have exhibited excess COVID-19 mortality in Sweden.[7] However, some notable differences have been found by immigrants' region of origin. For example, while immigrants from Finland and Chile

have shown moderately higher COVID-19 mortality risks than native-born Swedes, immigrants from other origins show relative risks approximately two (eg, Iran, Iraq), five (eg, Syria, Lebanon) and eight times (eg, Somalia) higher than that of natives.[7] This observation contrasts sharply with the mortality advantage seen among immigrants relative to native Swedes for most other causes, both before and during the pandemic.[7–10]

In the international literature, multiple hypotheses have been put forth to explain the excess COVID-19 mortality observed among immigrants and ethnic minorities, two categories which are commonly examined together. In the early months of the pandemic, explanations often focused on biological arguments (including the role of genetic predispositions and vitamin D levels), with politicians claiming that the 'virus does not discriminate' by social background.[11] However, as the pandemic continued, hypotheses involving social factors gained more attention, especially as the empirical evidence began to contradict previous reductive biological arguments. For one, excess COVID-19 mortality has been observed across a wide range of ethnic minority groups, calling into question the role of unique biological drivers.[5] Furthermore, COVID-19 has been found to have a disproportionately greater impact on lower socioeconomic groups,[12 13] illustrating the socially unequal effects of the pandemic.

While the research and political debate regarding the disproportionate impact of the pandemic on minorities is now generally discussed within a broader framework of social inequalities, empirical research on the mechanisms behind the observed inequalities is limited. A mechanistic understanding of the role of social inequalities in the pandemic is crucial for addressing the effects of the current crisis on already disadvantaged immigrant and ethnic minority groups, while equipping the public health field for potential future outbreaks.

### Theoretical foundation: social determinants of health

This study is rooted in the social determinants of health (SDH),[14] a theoretical framework that considers health to be a product of various social factors operating at different levels (individual, family, community and/or society) giving rise to social distributions of health and disease in the population. Although broad, the SDH framework lays the foundation for the study of health inequalities, focusing on the underlying social causes of manifested biological differences between groups.

The SDH framework becomes more informative in conjunction with other theoretical perspectives. For example, researchers should also account for the Developmental Origins of Health and Disease[15] and the life course[16] perspectives on health, which recognise that poor health is not only the result of the immediate social circumstances of an individual or group, but also of the individual and structural conditions that are present at birth or that accumulate over the lifespan, for example, socioeconomic deprivation or experiences of racism and discrimination. Within these frameworks, health inequalities can be conceptualised as the biological expression of unequal lives, which can persist within and across generations.

The above-mentioned frameworks lay the foundation for understanding how health inequalities are (re)produced by social factors operating through exposure and susceptibility.[17] This collective theory of health inequalities has been recently adapted to the context of the COVID-19 pandemic, both for the Swedish general population[12 18] and for ethnic minority groups internationally.[19] In this study, we will further adapt the theory to specifically consider inequalities in COVID-19 mortality among immigrants in Sweden.

Our theoretical expectation is that inequalities in COVID-19 mortality could arise from social factors operating through various points in the disease pathway (figure 1). First, social factors may result in differential exposure to the virus, that is, modifying the risk of being infected with SARS-CoV-2 in the first place. Social factors may also influence the development of the disease once the infection has occurred, with differential effects due to disease susceptibility (ie, socially patterned pre-existing health), as well as individual (ie, healthcare seeking) and structural (ie, healthcare provision) responses. These mechanisms, independently or in combination with one another, may, therefore, lead to differential risks of COVID-19 mortality.

More specifically, differential exposure to the virus[12 19] refers to how social conditions can place immigrants at greater risk of SARS-CoV-2 infection. For example, immigrants may be at risk by participating in front-line jobs or living in crowded households and densely populated or economically deprived areas with fewer opportunities to maintain a safe physical distance (eg, in neighbourhoods with small local stores and recreational facilities, fewer transport options for commuting and less access to outdoor space).

Differential effects via disease susceptibility[12 18] (also referred to as vulnerability[19]) relates to the biological capacity to fight against the virus once the infection has taken place. Immigrants may be at greater risk of mortality due to COVID-19 if they have more pre-existing health conditions or comorbidities, poorer nutritional status or immune response than native Swedes.[19] The concept of susceptibility has traditionally been used to refer to a genetic predisposition to certain diseases, which immediately removes from the debate the topic of inequalities (or inequities, ie, unfair and avoidable differences) to focus on fixed group-level variations. However, recent developments in epigenetics highlight the role of environmental conditions (including social inequalities) in shaping the expression of our genes and predisposing certain individuals and groups to poor health.[20] Such research allows us to situate the concept of susceptibility in the context of social inequalities.

Inequalities might also arise from differential effects of infection via individual and structural responses, that is, the inclination and capacity to seek and provide

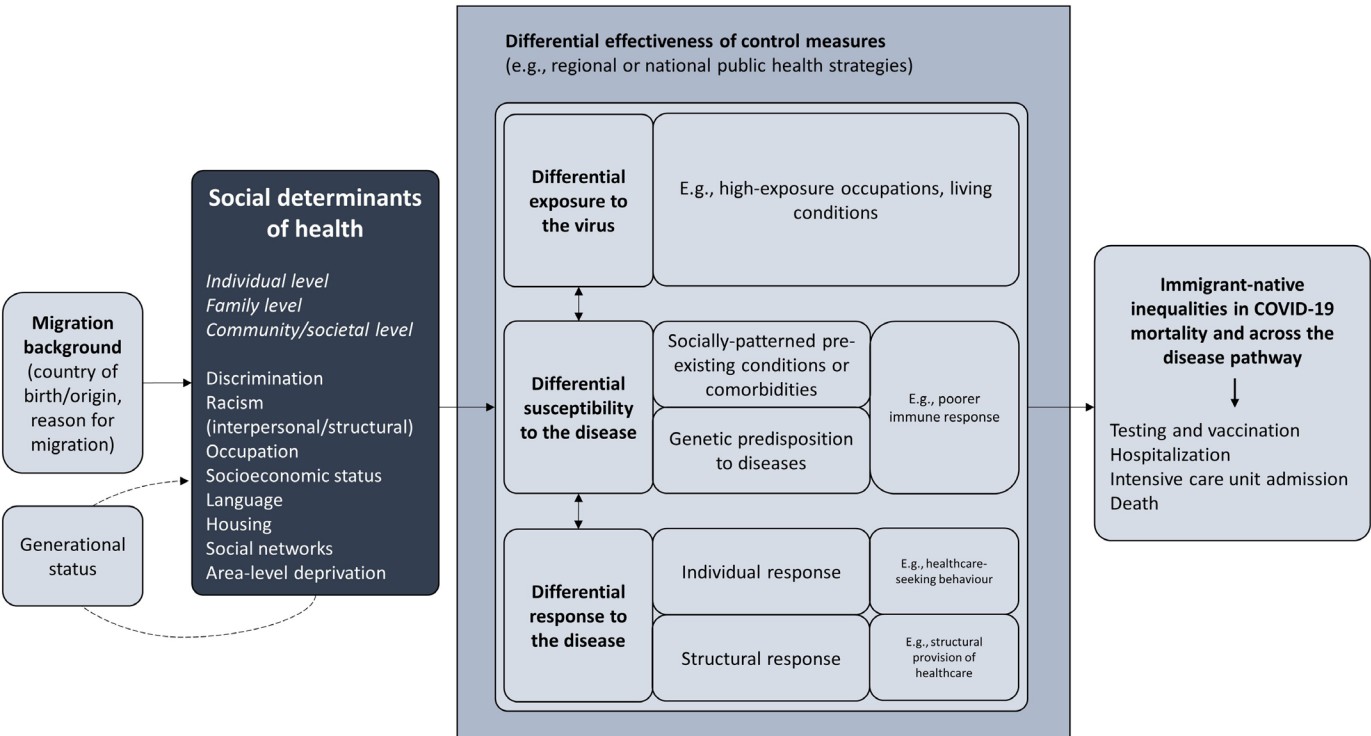

**Figure 1** Theoretical framework to investigate COVID-19 mortality among immigrants from a Social Determinants of Health perspective

necessary healthcare. Individual inclination to seek care may depend on one's trust in the healthcare system (depending on, eg, prior experiences of discrimination), while one's capacity may be dependent on the knowledge required to navigate the system (depending on, eg, language proficiency and health literacy). Furthermore, individual responses can to a large extent be conditioned by structural responses, or the capacity of the healthcare system to adapt to the demands of the pandemic while maintaining an equitable response across all social groups. Unlike disease susceptibility, which would be expected to remain stable throughout the course of the pandemic, structural response capacity may evolve and improve over time as the healthcare system adapts to the pandemic and more long-term solutions are introduced, such as the vaccination campaign. Yet, it is important to acknowledge the potential for continued inequalities in care provision rooted in underlying structural racism.

The above-mentioned mechanisms are not mutually exclusive.[19] Most adverse social factors are likely to have a joint effect on COVID-19 outcomes by operating across multiple mechanisms, not only resulting in an additive response on health but a multiplicative burden. For example, a greater risk of exposure to the virus in certain occupations might be exacerbated among individuals with high disease susceptibility who experience delayed or inadequate access to healthcare (ie, differential effects), resulting in more severe COVID-19 outcomes. In fact, given that social inequalities are implicated in all these mechanisms, it is actually expected that differential

exposure to the virus and differential effects of infection would be highly correlated.

Finally, it is important to acknowledge the potential for differential effectiveness of control measures, or the varied implementation and impact of regional or national public health strategies to tackle COVID-19.[19] By moderating the above-mentioned mechanisms, these control measures may have altered COVID-19 inequalities between immigrants and natives. Uniform control strategies affecting the total population (eg, lockdowns) may have reduced immigrant-native inequalities by minimising overall exposure to the virus, while less restrictive control measures (eg, work-from-home recommendations) may have selectively protected individuals with the capacity to follow measures, including individuals in non-service sectors. Information about control measures may have also been differentially accessible to natives and immigrants due to, for example, linguistic obstacles and lack of appropriate communication channels. Overall, the design and implementation of these control measures appears to have been shaped by pre-existing forms of institutional bias, including structural racism. Although we do not aim to estimate the differential impact of control measures in this study, we will take them into consideration when discussing the results.

### Aim and research questions

The purpose of this project is to study how elevated COVID-19 mortality risks among immigrants in Sweden are associated with social determinants acting through

differential exposure to infection (eg, being more likely to work in high-exposure occupations); and differential effects of infection resulting from socially patterned health conditions (ie, disease susceptibility) as well as social discrepancies in individual healthcare seeking and/or structural provision of healthcare (ie, individual or structural response).

Research question (RQ) 1: Are immigrants more likely to be exposed to SARS-CoV-2 infection than natives (differential exposure to the virus)?

RQ2: Are immigrants more likely to have socially patterned health conditions that predispose them to developing more severe versions of COVID-19 than natives (differential effects via disease susceptibility)?

RQ3: Are immigrants more likely to seek or receive healthcare at a later stage in their disease than natives (differential effects via individual or structural response)?

RQ4: How do the above-mentioned mechanisms interact to increase risks of COVID-19 mortality among different immigrant groups and compared with natives?

## METHODS AND ANALYSIS
### Study population and study period
The study population includes all adults (≥18 years old) registered in Sweden in the year before the start of the pandemic (31 January 2019) and individuals who immigrated to Sweden or turned 18 years old during follow-up.

Our analyses will primarily cover the period from 31 January 2020 (when the first confirmed COVID-19 case was registered in Sweden) to 31 December 2022, with updates depending on the development of the pandemic. In order to evaluate the specific effect of the pandemic on immigrant-native health inequalities, we will systematically estimate differences in all-cause mortality the year before the pandemic.

We will categorise the study period into two stages. The first period (from 31 January to 31 December 2020) reflects the time from the first recorded infection in Sweden until the start of the vaccination period (or the first date of vaccination data in Sweden, ie, 1 January 2021). The later period will be divided by considering different phases of access to vaccination by age and risk groups.

### Data
We will use multiple national registers linked by unique identification numbers[21]:
► Health registers, namely the Cause of Death (CDR),[22] Intensive Care (SIR), National Patient (NPR),[23] Prescription Drug,[24] SmiNet and National Vaccination Registers, with data on deaths, intensive care unit (ICU) visits, inpatient hospitalisations and outpatient visits, and drug prescriptions and dispensations (with dates, diagnoses for all), as well as COVID-19 PCR test results, individual vaccination dates, and sex and region for aggregated vaccination data, respectively;

► Socioeconomic registers, including the Longitudinal Integrated Database for Health Insurance and Labor Market Studies (LISA) and Longitudinal Database for Integration Studies, with annual data on socioeconomic indicators, social insurance benefits, municipality/county of residence, as well as immigrant-specific factors including country of birth, year of arrival in Sweden, and reason for migration.
► The Multigenerational Register,[25] to identify biological/adoptive family units.
► The Dwelling Register, to identify housing unit members, housing type, and dwelling characteristics.
► The Care and Services for the Elderly Register, to identify individuals living in elderly care homes.
► The Total Population Register, to allocate administrative (municipality, county and Small Areas for Market Statistics) and social (Demographic Statistical Areas) definitions of neighbourhoods.

We will use information from the above registers over different years. We will use socioeconomic information on the study population from 1990 (the year of establishment of LISA) to examine social patterns in underlying health status (from 2015) associated with severe COVID-19 (from 2019).

### COVID-19-related variables
This project will consider various COVID-19 outcomes along the disease pathway.

#### PCR tests
We will use data on positive COVID-19 PCR test dates and results. Depending on availability, we will examine individual-level or aggregate-level test data by sex and Swedish region.

#### Outpatient care
Data from the NPR will be used to capture outpatient visits coded with emergency/updated COVID-19 ICD codes: B34.2, U07.1 (diagnosed with laboratory tests), U07.2 (clinically diagnosed when laboratory tests are unreliable or unavailable); and dates to measure first contact with healthcare services for COVID-19 symptoms.

#### Hospitalisations
Data from the NPR will be used to capture hospitalisations coded with emergency/updated COVID-19 ICD codes, including dates of admission and release.

#### ICU admissions
Data from the SIR will be used to capture ICU admissions coded with emergency/updated COVID-19 ICD codes, including dates of admission and release.

#### Mortality
Using the CDR, we will capture deaths due to COVID-19, with dates and immediate/underlying causes of death by ICD codes. Given that COVID-19 mortality refers exclusively to confirmed cases, we will also evaluate excess mortality (ie, mortality levels exceeding those in the same

month in prepandemic years) and consider sensitivity analyses including deaths with a recent positive test but other CDR (eg, respiratory problems) that may be indicative of under-reported COVID-19 mortality.

### Delayed access or underutilisation of care

This will be proxied by different outcomes, including COVID-19 mortality outside hospitals; and time elapsed between testing/treatment phases. The latter includes time from testing positive for COVID-19 (see the PCR tests section) or first contact with healthcare services regarding COVID-19 symptoms (see the Outpatient care section) to hospitalisation; from COVID-19-related hospitalisation to ICU admission and from ICU admission to death.

### Immigrant-specific variables

All individuals born outside Sweden (approximately 20% of the Swedish population)[26] will be considered immigrants. They will be categorised by country/region of birth depending on the corresponding sample size and ethical considerations regarding identifiability. Duration of residence (<5, 5–10, >10 years), age at arrival (school-aged, young adult, older adult), intermarriage (native-native, native-immigrant, immigrant-native, immigrant-immigrant couples) and immigrant background (second generation) will be examined.

### Other relevant health variables

In order to measure health 5 years prior to a COVID-19 event, we will combine information from several health registers. We will primarily examine specific health conditions that have been shown to be associated with COVID-19 severity.[27] This includes chronic kidney disease (ICD-10 code: N18), diabetes (E08–E13), cardiovascular diseases (E78, G45–G46, I10–I13, I20–I26, I63–I66, I80–I82), neurological problems (G20, G21, G30, G35–G37), chronic respiratory diseases (J40–J47), tuberculosis (A15–A19), HIV (B20–B24), chronic liver disease (K70–K77), cancer (C00–C96) and thalassaemia or sickle-cell disorders (D56–D59). In addition, for the older adult population (65+ years), we will also calculate more general comorbidity measures, for example, the Charlson Comorbidity Index.

### Other relevant demographic and social variables

Across all studies, we will consider sex and age. Given that age is one of the most important predictors of COVID-19 prognosis, we will stratify our analyses by age groups, based on the availability of data for each age group and clinically meaningful patterns of COVID-19 prognosis across these age groups.

In order to investigate social factors that may affect the risk of exposure to infection for RQ1 and RQ4, we will assess variables at the individual level, including type of occupation (identifying high-exposure jobs using the O*NET online database[28] and employment type (including self-employed); household level, including type of dwelling (houses, apartments, care homes),

household size (individuals per square metre), number of working-age individuals (and their occupation) and presence of multiple generations in the household; and neighbourhood/residential level, including geographical proximity to family and the workplace, neighbourhood population density (residents per square kilometre) and immigrant density (proportion foreign-born residents).

The following variables will be carefully considered as confounders, moderators or mediators using directed acyclic graphs for all RQs: individual-level and area-level labour and disposable income, educational attainment, employment status, marital/cohabitation status, type and size of dwelling, and place of residence. Individual-level information will be used to account for area-level composition, while area-level information will be used to socioeconomically characterise the neighbourhood of residence.

### Study design and statistical analysis

This project will implement longitudinal study designs to effectively capture the timing of specific exposures and outcomes.

For RQ1 (differential exposure to the virus), we will compare the association of social factors hypothesised to influence risks of exposure and outcomes that confirm infection between immigrants and natives. We will apply multilevel models with three levels of analysis (individual, household, neighbourhood) in order to estimate specific (effect of individual-level and higher-level variables on individual outcomes) and general (effect of each level on the outcome) contextual effects, as well as to test cross-level interactions. We will also consider incorporating a fourth analytical level corresponding to the 21 Swedish regions, which are responsible for managing their own public health and healthcare responses to the pandemic beyond national recommendations.

We plan to implement a joint multilevel survival analysis in order to censor information by other causes of death or emigration.[29] In case of a lack of computational feasibility, we will run logistic hierarchical models. Individual variance will be interpreted using median HRs (survival models) or median ORs (logistic models) to evaluate the relative contribution of each level of analysis to the outcome. We will then compare model specifications to an empty model to estimate the proportional change in variance and evaluate the extent to which differences in the outcome are attributable to area level or compositional differences. Data on immigrant country/region of origin will be included as an individual-level variable (fixed characteristic) with native references. We will conduct additive and multiplicative interaction analyses to evaluate heterogeneity of exposure variables and outcomes between immigrants and natives.

For this question, two main concerns include positive reporting bias (ie, greater likelihood of taking a test following symptoms) and low testing rates among immigrants. A Swedish report found that immigrants from Africa had higher mortality than natives, but similar

risks of positive tests,[30] suggesting limited testing among specific immigrant groups who may still experience high COVID-19 infection rates. In order to overcome this challenge, we plan to supplement test data with outpatient, hospitalisation or ICU data coded for past COVID-19 infections to proxy a positive test result.

For RQ2 (differential effects via disease susceptibility), we will conduct two types of analyses. First, we will apply Cox regression to estimate time to first COVID-19 hospitalisation and death, using age as a timescale and right censoring by alternative (non-COVID-19 related) causes of death and emigration. Second, we will use mediation analyses to evaluate the extent to which native-immigrant differences in COVID-19 mortality are explained by pre-existing conditions at baseline (ie, at the start of the pandemic).

Given our interest in socially patterned conditions, we will also examine associations between the aforementioned health conditions and various social factors. In turn, we will be able to disentangle the extent to which pre-existing conditions involved in native-immigrant COVID-19 mortality differences are explained by social inequalities between these groups.

Underdiagnosis of pre-existing health conditions is one of the main challenges associated with this question. We thus plan to run subgroup analyses restricting the population to immigrants residing in Sweden for more than 10 years, since they may be more comparable to natives in terms of their healthcare-seeking and diagnosis rates.

For RQ3 (differential effects via disease response), we will use various approaches to determine whether immigrants have received less healthcare or at a later stage of the COVID-19 disease than natives. These include Poisson regression models to estimate relative risks of excess mortality between immigrants and native Swedes outside hospitals (ie, to capture unmet health needs) compared with before the pandemic. In addition, we will apply event history analyses with multiple time-scales[31] to evaluate transition rates between stages (testing/no-testing, hospitalisation/no-hospitalisation, ICU admission/no-ICU admission, death/no-death) while including age and calendar time (ie, pandemic waves). This approach will allow us to determine not only differential time elapsed between testing/treatment stages but also differential trajectories of testing/treatment between immigrants and natives. To disentangle the potentially confounding influence of differential susceptibility to severe disease (and potentially more rapid disease progression, see RQ2), we will control for the presence of pre-existing conditions.

Subgroup analyses will be conducted to distinguish differential individual and structural response mechanisms underlying native-immigrant care timelines. For individual response mechanisms, we will distinguish immigrant groups by duration of residence, age at arrival and occurrence of intermarriage to proxy individual obstacles to healthcare-seeking, for example, language proficiency or institutional awareness. To evaluate structural response mechanisms, we plan to assess whether inequalities in healthcare delay differ by time (within and across waves) and regions, during which resource availability (eg, ICU capacity) and other institutional factors (eg, vaccination progress) may have variably affected healthcare delivery to specific groups.

One challenge for this question is the potential measurement error affecting the timing and type of treatment for COVID-19 disease. For one, uncaptured delays in testing, not just treatment, among immigrants may underestimate the true time elapsed between infection and treatment. However, with postponed testing, we would in fact expect a shorter time between positive test and hospitalisation/ICU admission/death among immigrants than natives. Thus, whether natives or immigrants have a greater delay in receiving care could explain the primary underlying mechanism of postponed testing versus treatment.

More broadly speaking, attempts to disentangle the mechanisms from one another (eg, differential effects via disease susceptibility, individual response or structural response) may also be prone to bias. Although we plan to control for susceptibility through relevant pre-existing conditions, underdiagnosis may remain an issue (see RQ2). Similarly, it may be difficult to determine whether differential responses to the disease are due to individual healthcare-seeking behaviours or structural obstacles. We will thus rely on subgroup analyses to better understand these differing mechanisms, theorising that more 'integrated' immigrants, for example, those with longer residence or native partners, are less prone to underdiagnosis or healthcare-seeking obstacles, and thus more likely to be affected by structural obstacles such as inequitable resource allocation and healthcare delivery due to, for example, structural racism.

Finally, for RQ4 (all mechanisms), we will compile an overview of important theoretical and methodological considerations based on evidence from the first three RQs. For example, we will discuss how some social factors (eg, occupation) may be a common cause for different pathways (eg, differential exposure and differential effects via disease susceptibility). The explicit identification of different paths will be important for specific policy targets,[19] while the discussion regarding social causation will improve our understanding of COVID-19 inequalities. We plan to summarise the empirical studies of this project by considering potential interventions that can help prevent inequalities among immigrants in future pandemics.

Studies are planned for 2023–2025.

## Patient and public involvement

This project will not involve any patients or the public. However, we will attempt to communicate the findings of the study to the public through mass media.

## ETHICS AND DISSEMINATION

Although we handle personal information, individual consent is not required since the information is

anonymised by the agencies responsible for data protection, and none of the records will have personal identifiers attached. Statistics Sweden may also aggregate some information (such as country of birth) to ensure that individuals cannot be identified in relation to other variables (eg, education or place of residence). Nevertheless, since our dataset contains individual-level information and thus requires special provisions for its use, we applied for and were granted all necessary ethical permissions from the Swedish Ethical Review Authority to cover the specific RQs addressed by this project (Dnr 2022-0048-01).

The final outputs will primarily be disseminated as scientific articles published in open-access peer-reviewed international journals, as well as press releases and policy briefs.

## DISCUSSION

Most of the limitations in this study relate to problems of data availability and quality. First, we lack information on undocumented immigrants. Although we can identify undocumented (ie, without a personal identification number) immigrant deaths, individuals cannot be linked to register data (ie, social variables). Similarly, we are not able to identify ethnicity beyond the country of origin. Second, the project will study COVID-19 outcomes along the disease pathway, each of which may be affected by different sources of bias. Although we are not able to evaluate the role of the Swedish public health strategies specifically, we acknowledge that the lack of lockdown mandates, for example, can make our results less generalisable to other contexts, yet open opportunities for collaboration with research groups from other national contexts. The fact that Sweden did not have strong mandates (ie, lockdowns and face mask rules outside medical settings) will allow us to better examine the role of the different proposed mechanisms.

Despite these limitations, this study uses high-quality population registers that link multiple health and social outcomes before and after the start of the pandemic. The study will offer a thorough empirical evaluation of how social conditions shape group risks in the context of a public health crisis and can give rise to increased understanding of these mechanisms. Our findings will thus contribute knowledge to mitigating and potentially preventing such inequalities in future pandemics.

**Twitter** Sol Pia Juárez @501_Juarez and Helena Honkaniemi @H_Honkaniemi

**Acknowledgements** We would like to thank Dr. Anders Ledberg for his valuable insights on this protocol.

**Contributors** SPJ conceived the study. HH, SA, ED, MR, EM, SVK and AFC give critical feedback. SPJ drafted the manuscript with help of HH. SPJ and ED designed the visualisation of the theoretical framework. HH, SA, ED, MR, EM, SVK and AFC revised the manuscript. SPJ is the guarantor.

**Funding** This work was supported by the Swedish Council for Health, Working Life and Social welfare (FORTE), grant number 2021-00271. In addition, SPJ acknowledges funding from the Swedish Council for Health, Working Life and Social welfare (FORTE), grant number 2016-07128 and the Swedish Research Council (VR), gran number 2018-01825. SVK acknowledges funding from a NRS Senior Clinical Fellowship (SCAF/15/02), the Medical Research Council (MC_UU_00022/2) and the Scottish Government Chief Scientist Office (SPHSU17).

**Disclaimer** Funders of the study had no involvement in study design and will not be involved in any parts of the study, including data collection, data analysis, data interpretation, or writing of the report.

**Competing interests** None declared.

**Patient and public involvement** Patients and/or the public were not involved in the design, or conduct, or reporting, or dissemination plans of this research.

**Patient consent for publication** Not applicable.

**Provenance and peer review** Not commissioned; externally peer reviewed.

**ORCID iDs**
Sol Pia Juárez http://orcid.org/0000-0001-9086-7588
Helena Honkaniemi http://orcid.org/0000-0003-0800-0892
Siddartha Aradhya http://orcid.org/0000-0003-3748-6270
Enrico Debiasi http://orcid.org/0000-0001-5772-5510
Srinivasa Vittal Katikireddi http://orcid.org/0000-0001-6593-9092
Agneta F Cederström http://orcid.org/0000-0001-8866-7608
Eleonora Mussino http://orcid.org/0000-0002-5311-4277
Mikael Rostila http://orcid.org/0000-0002-6973-0381

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
