## [Reviewer comments · BMJ Open]

ARTICLE DETAILS

TITLE (PROVISIONAL)	Explaining COVID-19 mortality among immigrants in Sweden from a social determinants of health perspective (COVIS): protocol for a national register-based observational study
AUTHORS	Juárez , Sol Pia; Honkaniemi, Helena; Aradhya, Siddartha; Debiasi, Enrico; Katikireddi, Srinivasa; Cederström, Agneta; Mussino, Eleonora; Rostila, Mikael

VERSION 1 – REVIEW

REVIEWER	Gullon, Pedro Universidad de Alcala de Henares Facultad de Medicina y Ciencias de la Salud
REVIEW RETURNED	13-Feb-2023

GENERAL COMMENTS	Thank you for the opportunity to review the protocol entitled “Explaining COVID-19 mortality among immigrants in Sweden from a social determinant of health perspective (COVIS). Protocol for a population-register study”. The study is relevant, well-designed and its feasible under the strong public data that is available in Sweeden. All major limitations of the study are presented. Somo considerations: - I wonder if the authors want to analyze if inequalities change during the pandemic. Relative inequalities in socio-economic position have been found to be higher when lockdowns were lifted (e.g. in the Uk https://www.sciencedirect.com/science/article/pii/S0277953621007450?via%3Dihub and in Spain https://www.sciencedirect.com/science/article/pii/S1353829222000910?via%3Dihub). This might be specially interesting as the strategy of Sweeden has been different as other European countries.- In a similar direction, I wonder if regions or cities adopted strategies different than the national one, and if there is a way to test if these policies (or even structural determinants that are differential by region in Sweeden) can play a role in explaining the inequalities. I am no expert in the case of Sweeden, but some countries have regional differences in their systems of exposure to racism. Or even to think if neighborhoods are the relevant level for the multilevel or contextual variables at other levels might be relevant.- It is not clear to me where do you plan to get the data on testing, and I wonder if that includes not only PCR testing but other techniques that became more popular (antigen test). I don't know if this is a big issue if confirmed cases for mortality are only for those confirmed by PCR.
---

REVIEWER	Dowd, Jennifer University of Oxford, Leverhulme Centre for Demographic Science
REVIEW RETURNED	02-Mar-2023

GENERAL COMMENTS	This protocol proposes to use linked Swedish registry data to test the mechanisms through which immigrants in Sweden had higher COVID mortality during the pandemic. The impact of social
---

	determinants of health on COVID infection and mortality is an important topic, both for understanding the past and preparing for future crises. The data sources with individual linkages to occupation, etc, positive covid tests, hospitalization and death are enviable and provide a great opportunity to test some of these hypotheses. I'm certainly very pleased that these data are available and that this analysis will be done. A point of clarification- The early part of the text mentions "excess mortality" but in methods it is confirmed COVID deaths—it should be clear which will be analyzed as excess deaths require specific methods and estimation of baseline mortality. The main challenges I see in this proposed project are the feasibility of conducting a decomposition of different mechanisms and mediation analysis. RQ1 is straightforward and will provide good description of how individual social factors such as occupation, housing and family living arrangements, etc, were associated with confirmed infections. Since testing was not random however, if immigrants were much less likely to be tested this could lead to an underestimate of these associations. For RQ2, analyses of this type should certainly be done, but there are inherent challenges in trying to capture underlying vulnerabilities (for example something like the Charlson Index is very crude), so mediation analysis may not be super informative in the end. For example lack of complete mediation could be interpreted as either actual lack of mediation or just poor measurement of underlying health conditions, and it would be very hard to distinguish between the two. RQ3: I'm concerned that this measure of delayed access to care won't capture what it aims to capture. Presumably timing of testing also relates to social factors, so time from a positive test to interaction with the health system will be a combination of two different dynamics—waiting time from symptoms to testing and time from testing to seeking health care. In addition, even for those tested at roughly the same stage of infection, the time from test to hospitalization likely reflects severity and speed of disease progression itself, not just care seeking behavior. Thus, I'm concerned that this measure is not a good proxy for delays in care-seeking. Other things to look at descriptively might be the proportion of home deaths (which went up a lot in many countries), or other pre-pandemic measures associated with probability of attending health visits, etc? Q4: The challenges with the measurement in RQ3 make me less confident about the potential of the decomposition exercise described, as I don't think it will be possible to portion out mortality due to delayed care seeking in this way. Overall, this is an important project bringing together really rich individual and area level data. The descriptive aims will provide and important picture of factors associated with COVID infection, hospitalization, and death among immigrants in Sweden. I'm less optimistic about the ability to decompose the differential death rates neatly into these individual mechanisms, but the proposed project will push forward important knowledge nonetheless.
--	---

VERSION 1 – AUTHOR RESPONSE

Reviewer 1

Reviewer 1, #1: I wonder if the authors want to analyze if inequalities change during the pandemic. Relative inequalities in socio-economic position have been found to be higher when lockdowns were lifted (e.g. in the UK

<https://www.sciencedirect.com/science/article/pii/S0277953621007450?via%3Dihub> and in Spain <https://www.sciencedirect.com/science/article/pii/S1353829222000910?via%3Dihub>). This might be specially interesting as the strategy of Sweden has been different as other European countries.

AUTHORS: Thank you for the comment. Given that Sweden did not experience a lockdown like other countries, we have so far considered two main stages (pre- and post-vaccination periods) but we are aware that our results, particularly within the first phase, could be certainly relevant to compare with countries that have adopted a different public health strategy. In fact, we are interested in establishing collaborations with other research groups to be able to better evaluate the impact of different approaches on immigrant-native COVID-19 inequalities. However, since these collaborations are only hypothetical so far, we did not want to add a specific research question. That being said, we have now better acknowledged this component in the background (see below) and in our model (see Figure 1), as well as addressed this as a limitation (see below).

In the background: “Finally, it is important to acknowledge the potential for differential effectiveness of control measures, or the varied implementation and impact of regional or national public health strategies to tackle COVID-19. By moderating the above-mentioned mechanisms, these control measures may have altered COVID-19 inequalities between immigrants and natives. Uniform control strategies affecting the total population (e.g., lockdowns) may have reduced immigrant-native inequalities by minimizing overall exposure to the virus, while less restrictive control measures (e.g., work-from-home recommendations) may have selectively protected individuals with the capacity to follow measures, including individuals in non-service sectors. Information about control measures may have also been differentially accessible to natives and immigrants due to e.g., linguistic obstacles and lack of appropriate communication channels. Overall, the design and implementation of these control measures appears to have been shaped by pre-existing forms of institutional bias, including structural racism. Although we do not aim to estimate the differential impact of control measures in this study, we will take them into consideration when discussing the results.” (page 6)

In the limitation: “Although we are not able to evaluate the role of the Swedish public health strategies specifically, we acknowledge that the lack of lockdown mandates, for example, can make our results less generalizable to other contexts, yet open opportunities for collaboration with research groups from other national contexts. The fact that Sweden did not have strong mandates (i.e., lockdowns and use of face masks outside medical settings) will allow us to better examine the role of the different proposed mechanisms.” (page 12)

Reviewer 1, #2: In a similar direction, I wonder if regions or cities adopted strategies different than the national one, and if there is a way to test if these policies (or even structural determinants that are differential by region in Sweden) can play a role in explaining the inequalities. I am no expert in the case of Sweden, but some countries have regional differences in their systems of exposure to racism. Or even to think if neighborhoods are the relevant level for the multilevel or contextual variables at other levels might be relevant.

AUTHORS: This is another very good suggestion that we believe fits nicely with our research question #3. We have not expanded much on this point in the project description on purpose because, although regional differences are expected (as healthcare in Sweden is decentralized) we have not been able to identify any clear difference in the implementation of the national recommendations that, for example, allows for a quasi-experimental design. However, for RQ1 regarding potentially differential exposure to the virus, we were planning to acknowledge possible

regional variation in the models (i.e., region-fixed effects) if it is not possible to run a four-level multilevel model due to computational challenges (see below). Structural response differences will also be considered by region for RQ3.

“We will also consider incorporating a fourth analytical level corresponding to the 21 Swedish regions, which are responsible for managing their own public health and healthcare responses to the pandemic beyond national recommendations.” (page 10)

“To evaluate structural response mechanisms, we plan to assess whether inequalities in healthcare delay differ by time (within and across waves) and regions, during which resource availability (e.g., ICU capacity) and other institutional factors (e.g., vaccination progress) may have variably affected specific groups’ ability to seek immediate treatment.” (page 11)

Reviewer 1, #3: It is not clear to me where do you plan to get the data on testing, and I wonder if that includes not only PCR testing but other techniques that became more popular (antigen test). I don’t know if this is a big issue if confirmed cases for mortality are only for those confirmed by PCR.

AUTHORS: Unfortunately, information is only available for PCR testing. To evaluate the role of higher exposure in explaining higher mortality among migrants, we will have to approximate infection considering all data available (i.e., COVID-related hospitalizations, ICU admissions, mortality and PCR tests). Although this is the best we can do in the absence of a seroprevalence study, we are fully aware that this is a very imperfect measure of (only symptomatic) infection that combines severe forms of the disease and positive testing information, the latter of which is subject to a high degree of underreporting. We will be very careful in describing our data limitations and the scope of our findings in our studies.

“For this question, two main concerns include positive reporting bias (i.e., greater likelihood of taking a test following symptoms) and low testing rates among immigrants. A Swedish report found that immigrants from Africa had higher mortality than natives, but similar risks of positive tests,³⁰ suggesting limited testing among specific immigrant groups who may still experience high COVID-19 infection rates. In order to overcome this challenge, we plan to supplement test data with outpatient, hospitalization or ICU data coded for past COVID-19 infections to proxy a positive test result.” (page 10)

We include confirmed cases of COVID mortality by PCR (U07.1) or via clinical diagnosis when PCR is unreliable/unavailable (U07.2). Autopsies have not been conducted in Sweden. For this reason, we plan to accompany our analyses of COVID-19 specific mortality with excess mortality during the pandemic. This could help us to determine whether underreporting of COVID deaths is likely.

“Outpatient care: Data from the NPR will be used to capture outpatient visits coded with emergency/updated COVID-19 ICD codes: B34.2, U07.1 (diagnosed with laboratory tests), U07.2 (clinically diagnosed when laboratory tests are unreliable or unavailable); and dates to measure first contact with healthcare services for COVID-19 symptoms.” (page 8)

Reviewer 2

Reviewer 2, #1: A point of clarification- The early part of the text mentions “excess mortality” but in methods it is confirmed COVID deaths—it should be clear which will be analyzed as excess deaths require specific methods and estimation of baseline mortality.

AUTHORS: Thank you for noticing this. We realize now that we have mistakenly used this term as a synonym for a ‘disproportionately higher COVID-19 mortality among immigrants’. We have tried to be more careful now and also more explicit about when we plan to examine excess mortality (i.e., higher mortality levels compared to the same month from years before the pandemic).

Reviewer 2, #2: The main challenges I see in this proposed project are the feasibility of conducting a decomposition of different mechanisms and mediation analysis. RQ1 is straightforward and will provide good description of how individual social factors such as occupation, housing and family living

arrangements, etc, were associated with confirmed infections. Since testing was not random however, if immigrants were much less likely to be tested this could lead to an underestimate of these associations.

AUTHORS:

Please see our response about the decomposition analysis in line with your comment #5 below.

In relation to the underestimation of testing among immigrants, we will certainly have to be careful when interpreting the results and, of course, acknowledge our data limitations. We also hope we can compensate underreported testing among immigrants by considering infection, using not only testing (through PCR) but all available COVID-related outcomes (such as severe forms of the disease, where immigrants are overrepresented). In fact, although our project focuses on COVID mortality, we are aware that our studies need to also examine other COVID-related outcomes to better interpret our findings. In our submission, we mentioned this strategy under the “Limitations” section. However, for clarity we now discuss the limitations following each of the research questions in the “Study design and statistical analysis” section, including how we plan to address them.

For RQ1: “For this question, two main concerns include positive reporting bias (i.e., greater likelihood of taking a test following symptoms) and low testing rates among immigrants. A Swedish report found that immigrants from Africa had higher mortality than natives, but similar risks of positive tests,³⁰ suggesting limited testing among specific immigrant groups who may still experience high COVID-19 infection rates. In order to overcome this challenge, we plan to supplement test data with outpatient, hospitalization or ICU data coded for past COVID-19 infections to proxy a positive test result.” (page 10)

For RQ3: “One challenge for this question is the potential measurement error affecting the timing and type of treatment for COVID-19 disease. For one, uncaptured delays in testing, not just treatment, among immigrants may underestimate the true time elapsed between infection and treatment. However, with postponed testing, we would in fact expect a shorter time between positive test and hospitalization/death among immigrants than natives. Thus, whether natives or immigrants have a greater delay in receiving care could explain the primary underlying mechanism of postponed testing versus treatment.

More broadly speaking, attempts to disentangle the mechanisms from one another (e.g., differential effects via disease susceptibility, individual response or structural response) may also be prone to bias. Although we plan to control for susceptibility through relevant pre-existing conditions, underdiagnosis may remain an issue (see RQ2). Similarly, it may be difficult to determine whether differential responses to the disease are due to individual healthcare-seeking behaviors or structural obstacles. We will thus rely on sub-group analyses to better understand these differing mechanisms, theorizing that more ‘integrated’ immigrants, e.g., those with longer residence or native partners, are less prone to underdiagnosis or healthcare-seeking obstacles, and thus more likely to be affected by structural obstacles such as inequitable resource allocation due to, e.g., structural racism.” (page 12)

Reviewer 2, #3: For RQ2, analyses of this type should certainly be done, but there are inherent challenges in trying to capture underlying vulnerabilities (for example something like the Charlson Index is very crude), so mediation analysis may not be super informative in the end. For example lack of complete mediation could be interpreted as either actual lack of mediation or just poor measurement of underlying health conditions, and it would be very hard to distinguish between the two.

AUTHORS: Thank you for highlighting this caveat. We agree that estimating underlying conditions is not an easy task. In fact, we would like to highlight that most of our research questions probably require more than one strategy to evaluate vulnerabilities (i.e., more than one paper). Counting morbidities using the Charlson Index among the 65+ year old population is one of multiple strategies, and likely not the best strategy for the younger population. For RQ2, we plan to evaluate morbidities

that have previously been associated with severe COVID-19 disease in the literature, and to determine the extent to which they explain the higher risk of COVID-19 mortality among immigrants. We have tried to be more explicit about this in the current version. In any case, we acknowledge that a detailed description of the limitations of our data and a careful interpretation of our findings is necessary in all our studies.

“In order to measure health five years prior to a COVID-19 event, we will combine information from several health registers. We will primarily examine specific health conditions that have been shown to be associated with COVID-19 severity.²⁷ This includes: chronic kidney disease (ICD-10 code: N18), diabetes (E08-E13), cardiovascular diseases (E78, G45-G46, I10-I13, I20-I26, I63-I66, I80-I82), neurological problems (G20, G21, G30, G35-G37), chronic respiratory diseases (J40-J47), tuberculosis (A15-A19), HIV (B20-B24), chronic liver disease (K70-K77), cancer (C00-C96) and thalassemia or sickle-cell disorders (D56-D59). In addition, for the older adult population (65+ years old) we will also calculate, e.g., the Charlson Comorbidity Index.” (page 9)

Reviewer 2, #4: RQ3: I'm concerned that this measure of delayed access to care won't capture what it aims to capture. Presumably timing of testing also relates to social factors, so time from a positive test to interaction with the health system will be a combination of two different dynamics—waiting time from symptoms to testing and time from testing to seeking health care. In addition, even for those tested at roughly the same stage of infection, the time from test to hospitalization likely reflects severity and speed of disease progression itself, not just care seeking behavior. Thus, I'm concerned that this measure is not a good proxy for delays in care-seeking. Other things to look at descriptively might be the proportion of home deaths (which went up a lot in many countries), or other pre-pandemic measures associated with probability of attending health visits, etc?

AUTHORS: Thank you for your input on this research question. We completely understand the reviewer's concern regarding the interpretation of the results and we would like to acknowledge that we are aware of this challenge. We expect that immigrants experience a systematic diagnostic delay across all outcomes and we base this expectation on our pre-pandemic knowledge, where immigrants (especially newcomers) experience language barriers and problems navigating the healthcare system. In the context of a pandemic, where new routines were implemented and, at least in the beginning, only in Swedish, we believe that the barriers could only be worse for immigrants. That being said, we understand the reviewer's concern on how to interpret timing between stages because, indeed, it would be difficult to isolate the health effects of delay to care and the speed of the disease progression. This issue will be particularly relevant if we conclude that immigrants in Sweden are indeed more vulnerable than natives. To our knowledge, this hypothesis is not conclusive in the international literature and certainly not in Sweden. For one, immigrants have been shown to have a mortality advantage before the pandemic. Furthermore, the (very few) studies that have accounted for (some) pre-existing conditions did not observe that these conditions explained a substantial part of the inequalities in COVID-19 outcomes between immigrants and Swedes. In addition, in a recent study we explored pre-pandemic differences in causes of hospitalization associated with higher risks of COVID-19 severity between immigrants and natives in Sweden. We concluded that immigrants were generally at lower risk of experiencing hospitalizations for most of these problems compared to natives, with the exception of some low-prevalence causes. The conclusion of this paper serves as an example that further studies are needed to prove that vulnerability actually plays a role in explaining the higher risk of COVID-19 mortality among immigrants.

That being said, in order to address issues of testing and treatment timing, as well as disentangle delay to care and speed of disease progression, we have now suggested several sub-analyses specifically for this purpose..

“One challenge for this question is the potential measurement error affecting the timing and type of treatment for COVID-19 disease. For one, uncaptured delays in testing, not just treatment, among immigrants may underestimate the true time elapsed between infection and treatment. However, with postponed testing, we would in fact expect a shorter time between positive test and

hospitalization/death among immigrants than natives. Thus, whether natives or immigrants have a greater delay in receiving care could explain the primary underlying mechanism of postponed testing versus treatment.

More broadly speaking, attempts to disentangle the mechanisms from one another (e.g., differential effects via disease susceptibility, individual response or structural response) may also be prone to bias. Although we plan to control for susceptibility through relevant pre-existing conditions, underdiagnosis may remain an issue (see RQ2). Similarly, it may be difficult to determine whether differential responses to the disease are due to individual healthcare-seeking behaviors or structural obstacles. We will thus rely on sub-group analyses to better understand these differing mechanisms, theorizing that more ‘integrated’ immigrants, e.g., those with longer residence or native partners, are less prone to underdiagnosis or healthcare-seeking obstacles, and thus more likely to be affected by structural obstacles such as inequitable resource allocation due to, e.g., structural racism.” (page 12) Please note that we have also adjusted the modelling strategy in relation to this question. We have decided to replace Cox regression models with an event history analysis with multiple time-scales. Although the two have similar properties, the newer approach would permit us to simultaneously use age at the baseline while being able to account for the time between different stages of the disease path (PCR test, outpatient care, hospitalization, ICU admission and death). In addition, this model will allow us to identify whether and when there is a differential transition between stages for immigrants and natives across the disease pathway.

“For RQ3 (differential effects via disease response), we will use various approaches to determine whether immigrants have received healthcare at a later stage of the COVID-19 disease. These include Poisson regression models to estimate relative risks of excess mortality between immigrants and native Swedes outside hospitals (i.e., to capture unmet health needs) compared to before the pandemic. In addition, we will apply event history analyses with multiple time-scales³¹ to evaluate transition rates between stages (testing/no-testing, hospitalization/no-hospitalization, ICU admission/no-ICU admission, death/no-death) while including age and calendar time (i.e., pandemic waves). This approach will allow us to determine not only differential time elapsed between testing/treatment stages but also differential trajectories of treatment between immigrants and natives. To disentangle the potentially confounding influence of differential susceptibility to severe disease (and potentially more rapid disease progression, see RQ2), we will control for the presence of pre-existing conditions.” (page 11)

Reviewer 2, #5: Q4: The challenges with the measurement in RQ3 make me less confident about the potential of the decomposition exercise described, as I don’t think it will be possible to portion out mortality due to delayed care seeking in this way.

AUTHORS: We understand the reviewer’s concern regarding the decomposition method and have decided to adjust our analytical strategy accordingly. In fact, as formulated in the research question, our aim is to consider the interaction between these different mechanisms rather than quantify which one has a greater role in explaining higher COVID-19 mortality among immigrants. As we highlighted in the introduction, these mechanisms are not mutually exclusive and they can also be tackled with different interventions.

Given that this research question strongly depends on our previous findings, it is difficult to propose a coherent analytical strategy at this point. Furthermore, we believe that this final question likely needs to be tackled from a theoretical perspective as well, as an overall summary of our previous empirical studies. This summary could better shed light on the interventions that are needed to prevent and mitigate the adverse consequences of future pandemics on the immigrant population.

It now reads:

“Finally, for RQ4 (all mechanisms), we will compile an overview of important theoretical and methodological considerations based on evidence from the first three RQs. For example, we will discuss how some social factors (e.g., occupation) may be a common cause for different pathways (e.g., differential exposure to the virus and differential effects via disease susceptibility). The explicit

identification of different paths will be important for specific policy targets,¹⁹ while the discussion regarding social causation will improve our understanding of COVID-19 inequalities. As part of this last question, we plan to summarize the empirical studies of this project by considering potential interventions that can help prevent inequalities among immigrants in future pandemics.” (page 12)

Reviewer 2, #6: Overall, this is an important project bringing together really rich individual and area level data. The descriptive aims will provide an important picture of factors associated with COVID infection, hospitalization, and death among immigrants in Sweden. I'm less optimistic about the ability to decompose the differential death rates neatly into these individual mechanisms, but the proposed project will push forward important knowledge nonetheless.

AUTHORS: Thank very much for your thorough evaluation of our protocol. Your concerns have really helped us to better design our future work and we hope this is apparent in the new version of our manuscript.